# A Multichannel MAC Protocol without Coordination or Prior Information for Directional Flying Ad hoc Networks

**Shijie Liang** [1,2], **Haitao Zhao** [2,*], **Jiao Zhang** [2], **Haijun Wang** [2], **Jibo Wei** [2] and **Junfang Wang** [1]

1   The 54th Research Institute of China Electronics Technology Group Corporation, Shijiazhuang 050081, China; shijieliang21@163.com (S.L.); jfwang63@163.com (J.W.)
2   College of Electronic Science, National University of Defense Technology, Changsha 410073, China; zhangjiao16@nudt.edu.cn (J.Z.); haijunwang14@nudt.edu.cn (H.W.); wjbhw@nudt.edu.cn (J.W.)
*   Correspondence: haitaozhao@nudt.edu.cn; Tel.: +86-1552-115-7303

**Abstract:** Achieving neighbor discovery for a directional flying ad hoc network (FANET) with multiple channels poses challenges for media access control (MAC) protocol design, as it requires simultaneous main lobe and channel rendezvous while dealing with the high UAV mobility. In order to achieve fast neighbor discovery for initial access without coordination or prior information, we first establish the theoretical supremum for the directional main lobe. Then, to achieve the supremum, we introduce the BR-DA and BR-DA-FANET algorithms to respectively establish the supremum on rendezvous between a pair of UAVs' main lobes and rendezvous of main lobes for all UAVs in the FANET. To further accelerate the neighbor discovery process, we propose the neighbor discovery with location prediction protocol (ND-LP) and the avoiding communication interruption with location prediction (ACI-LP) protocol. ND-LP enables quick main lobe rendezvous and channel rendezvous, while ACI-LP enables beam tracking and channel rendezvous together with the avoidance of communication interruptions. The simulation results demonstrate that the proposed protocols outperform the state-of-the-art works in terms of neighbor discovery delay.

**Keywords:** beam rendezvous; channel rendezvous; directional antenna; flying ad hoc network (FANET); medium access control (MAC); neighbor discovery





## 1. Introduction

In recent years, unmanned aerial vehicles (UAVs) have garnered considerable attention due to their ability to perform tasks that are challenging or hazardous for humans, including reconnaissance, surveillance, and search and rescue [1–3]. Compared to a single UAV, multiple cooperative UAVs can provide broader coverage, greater flexibility, and robustness through versatility [4–6]. Therefore, the flying ad hoc network (FANET) has become a hot topic as it provides an effective real-time communication solution for UAVs' collaboration.

However, the UAVs in FANETs usually encounter several challenges, such as limited communication resources, intensive contention from neighboring UAVs, and even external interference, which can hinder meeting network capacity and delay communication requirements. In order to increase network capacity, the usage of multiple channels [7,8] and directional antennas [8–11] are essential for FANETs. Due to their advantages in terms of high gain and focused beamforming capabilities, directional antennas can cover a larger communication range compared to omnidirectional antennas [12] and are well-suited for jamming environments [13]. At the same time, UAVs equipped with narrow beam angle directional antennas pose design difficulties for the media access control (MAC) protocol. Neighbor discovery for a MAC protocol must fulfill the requirement that any pair of neighbor UAVs steers their main lobe toward each other to complete main lobe rendezvous and operate in different modes (reception and transmission mode) simultaneously within a limited time. Additionally, the MAC protocol must guarantee that main lobe rendezvous interruption and data transmission interruption are avoided due to the high mobility of UAVs.

Last but not least, the design of the MAC protocol needs to complete simultaneous channel rendezvous [14] and main lobe rendezvous [15] in a multichannel FANET with directional antennas. Many studies have investigated neighbor discovery in directional networks. Neighbor discovery for directional FANETs can be generally classified into two types: one is to achieve neighbor discovery with the help of omnidirectional antennas [16–19], while the other adopts directional antennas directly to achieve neighbor discovery [15,20–26]. The communication range of omnidirectional antennas is commonly smaller than that of directional antennas. Therefore, the neighbor set discovered through the former is a subset of that of the latter. There are two ways to extend the communication range with omnidirectional antennas. One is to exploit multihops [27], whereas the increase of multihops could introduce negative impacts such as large delays, additional interruptions, and a higher bit error rate (BER) [28,29]. The other method is dual-band wireless networks [30]. According to the Friis transmission equation [31], the high loss of the high-frequency band is offset by the gain of the directional antenna so that the omnidirectional antenna communication range is the same as that of the directional antenna. However, this approach requires coordination between different frequency bands.

Therefore, many studies for neighbor discovery focus on using directional antennas directly. In [20], an adaptive directional antenna protocol for terahertz networks (ADAPT) was proposed to achieve high throughput. However, this protocol was designed for the directional networks with centralized architecture, which is not suitable for FANETs. There are two methods of neighbor discovery for FANETs with directional antennas: probabilistic [21–23] and deterministic protocols [15,24–26]. The disadvantage of probabilistic protocols is that the discovery delay cannot be guaranteed, which may lead to extreme delays for two neighbor UAVs to discover each other. Deterministic protocols, on the other hand, achieve neighbor discovery with a bounded delay. In [25,26], an iterative search was proposed to decrease the delay of neighbor discovery. However, an iterative search is not suitable for environments that are characterized by jamming phenomena which increase the misdetection probability [32]. In [15,24], predefined scan sequences and modes were exploited to discover neighbors for directional ad hoc networks without coordination or prior information.

Taking advantage of multiple channels, the capacity of a network with directional antennas can be further increased. However, achieving channel rendezvous and main lobe rendezvous simultaneously is not trivial, and hence, poses great challenges for MAC protocol design. In [16,18], two MAC protocols were proposed for networks with channel rendezvous and main lobe rendezvous to achieve a large capacity network with the help of omnidirectional antennas.

In summary, previous studies for neighbor discovery in directional FANETs that use omnidirectional antennas face the problem of being unable to discover all neighboring UAVs due to the low gain of the omnidirectional antenna. However, the neighbor discovery protocols that rely solely on directional antennas do not reach the theoretical supremum and take into account channel rendezvous for directional FANETs with multiple channels. To address these problems, a multichannel MAC protocol with directional antennas for FANETs (FA-MMAC-DA) is proposed to achieve quick simultaneous channel rendezvous and main lobe rendezvous [15]. The FA-MMAC-DA can accomplish neighbor discovery for all UAVs in a network and avoid communication interruption. The FA-MMAC-DA consists of the BR-DA, BR-DA-FANET, neighbor discovery with location prediction (ND-LP), and avoiding communication interruption with location prediction (ACI-LP) protocols. The main contributions are summarized as follows:

- We present a system model for multichannel FANETs with double-directional antennas. One directional antenna is used for neighbor discovery in the control channel, and the other is exploited to transmit data in data channels.
- We establish the theoretical supremum for neighbor discovery during initial access [30] without coordination or prior information and propose a blind rendezvous algorithm to achieve the theoretical supremum. We extend the blind rendezvous algorithm to

neighbor the discovery protocols BR-DA and BR-DA-FANET to achieve neighbor discovery for a pair of nodes and the entire network, respectively, in scenarios without prior information or coordination.

- In order to further decrease delay, we propose the neighbor discovery with location prediction (ND-LP) protocol after initial access neighbor discovery and the avoiding communication interruption with location prediction (ACI-LP) protocol during the data transmission process. The predicted location is utilized for quick main lobe rendezvous and channel rendezvous in the ND-LP protocol and beam tracking together with channel rendezvous in the ACI-LP protocol.

The remainder of this paper is organized as follows: Section 2 introduces the system model and the problem definition of this research. Section 3 describes the details of our proposed algorithm, the MAC protocol, and theoretical analysis of their performance. Section 4 evaluates the FA-MMAC-DA protocol through extensive simulations and analysis. Finally, Section 5 concludes this research. In the paper, abbreviations are presented in Table 1, while the variable and parameter descriptions are provided in Table 2.

**Table 1.** Summary of abbreviations.

| Abbreviation | Definition | Abbreviation | Definition |
|---|---|---|---|
| MAC | Media Access Control | BER | Bit Error Rate |
| ND-LP | Neighbor discovery with location prediction | ACI-LP | Avoiding communication interruption with location prediction |
| GPS | Global positioning system | SDR | Software-defined radio |
| FA-MMAC-DA | Multichannel MAC protocol with directional antennas for a FANET | WCDMR | Worst-case-delay-to-main-lobe-rendezvous |

**Table 2.** Summary of variable and parameter.

| Symbol | Definition | Symbol | Definition |
|---|---|---|---|
| $\theta$ | The main lobe angle of the directional antenna | $l_1$ | The length of the segment of 1's in a specific control sequence |
| $N$ | The number of sectors | $l_2$ | The length of the UAV's unique ID |
| $\theta_S$ | The width of the main lobe for the switched beam antenna | $U$ | The maximum number of UAVs in a FANET |
| $\theta_P$ | The main lobe angle of the phased array antenna | $(x_i, y_i)$ | The location of the $i$-th UAV |
| $\varphi$ | The direction of communication | $v_i$ | The speed of the $i$-th UAV |
| $\psi$ | The direction of the main lobe boresight | $\zeta_i$ | The angle between the movement direction of the $i$-th UAV and the positive direction of the $X$ axis |
| $\boldsymbol{u}$ | The indexes of sectors | $t_i$ | The time that information is transmitted in by the $i$-th UAV |
| $P_b$ | The UAV $b'$ sector that UAV $a$ is located in | $(x_T, y_T)$ | The transmitter location |
| $P_a$ | The UAV $a'$ sector that UAV $b$ is located in | $(x_R, y_R)$ | The receiver location |
| $I_a$ | The sector that UAV $a$ points towards in the initial state | $D_{T,R}$ | The distance between the transmitter and receiver |
| $\boldsymbol{S}$ | The antenna scan sequence | $D_{max}$ | The maximum communication distance |
| $S_a$ | The antenna scan sequence for UAV $a$ | $T_{CS}$ | The delay for sensing and judging channels' availability |

**Table 2.** *Cont.*

| Symbol | Definition | Symbol | Definition |
|---|---|---|---|
| $T_{rs}$ | The worst-case discovery delay for a pair of UAVs | $T_{TS2-RTS}$ | The delay of TS2-RTS |
| $M_T T$ | The time duration for the transmitter UAV | $T_{TS2-CTS}$ | The delay of TS2-CTS |
| $M_R T$ | The minimal slot duration to exchange discovery beacons between the pair of UAVs | $T_{Inf}$ | The duration of the preparation-to-transmit information |
| $l_0$ | The length of the segment of $0's$ in a specific control sequence | $\Delta t$ | The sum of the duration of the data packet and $T_{Inf}$ |

## 2. System Model and Problem Definition

In this paper, we consider a scenario where each UAV is equipped with two directional antennas: a switched beam antenna and a phased array antenna. The switching delay for antenna sectors is negligible. We assume that the software-defined radio (SDR) technology is adopted so that all UAVs can work on different channels and switch channels flexibly. We assume that there is no interference among channels. The network and antenna models are described in Section 2.1, the multichannel model of the directional FANET is described in Section 2.2, and the formulation of the problem based on the model is found in Section 2.3.

### 2.1. Network Model and Antenna Model

We consider a FANET deployed over a large area. Each UAV in the FANET is equipped with two independent transceiver terminals with directional antennas. One transceiver is responsible for exchanging control packets and is denoted as TS1. The other transceiver transmits data packets and is denoted as TS2. Both TS1 and TS2 work in half duplex mode. To reduce costs, we consider using switched beam and phased array antennas [33] for the directional antennas on TS1 and TS2, respectively. The directional antenna has a main lobe angle $\theta$.

Switched beam antenna model: The angle of each sector is denoted by $\theta_S$ ($0 < \theta_S \leq 2\pi$) which is also the width of the main lobe and sidelobes, and the directional antenna pattern consists of $N-1$ sidelobes and a main lobe as illustrated in Figure 1. The relationship between $\theta_S$ and $N$ is expressed as:

$$\theta_S = \frac{2\pi}{N} \tag{1}$$

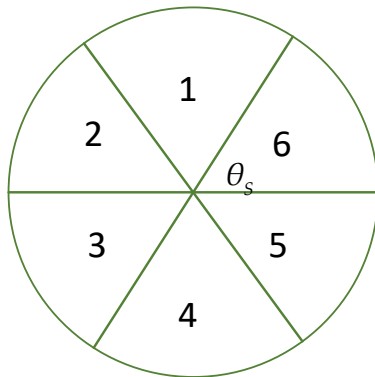

**Figure 1.** Switched beam antenna model.

The communication coverage is divided into $N$ non-overlapping sectors, with sector indexes ranging from 1 to $N$ in counterclockwise order. The main lobe needs to scan $N$ times to cover all sectors at a minimum. As an example in Figure 1, the angle of the sector is $\frac{\pi}{3}$, and therefore the main lobe of the directional antenna can scan all directions by 6 scans. The indexes of the sectors are $\boldsymbol{u} = \{1, 2, 3, 4, 5, 6\}$.

Phased array antenna model: Compared to the *N* fixed directions of the switched beam antenna, the main lobe of the phased array antenna can point in any direction. We denote $\theta_P$ as the main lobe angle of TS2. Note that $\theta_S$ is usually bigger than $\theta_P$ to accelerate neighbor discovery. When the main lobe rendezvous is achieved for TS2, the boresight of the transmitter and receiver overlap, which is shown in Figure 2. In Figure 2, A and B represent the transmitter and receiver, respectively. C and D are the intersection points of the two beam edges. Hence, $\angle CAB$, $\angle DAB$, $\angle CBA$ and $\angle DBA$ are $\frac{\theta_P}{2}$.

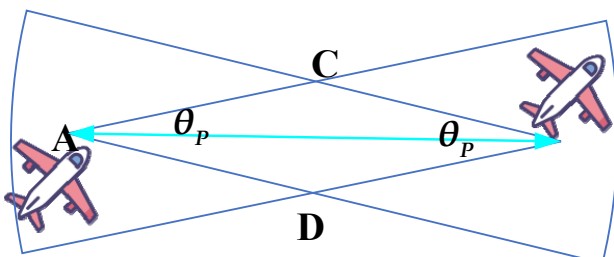

**Figure 2.** Phased array antenna.

Let $\varphi$ denote the direction of communication and $\psi$ denote the direction of main lobe boresight, and we adopt the following antenna model [23,34]:

$$g(\varphi) = \begin{cases} G(\theta), |\varphi - \psi| \leq \frac{\theta}{2} \\ 0, otherwise \end{cases}, \tag{2}$$

### 2.2. Multichannel Directional FANET Model

There is one channel for the control frame and several channels for the data frame. We define a directional multichannel network structure, as shown in Figure 3. In this structure, the time is divided into fixed slots, and each slot is further divided into several channels by frequency. Note that the direction can be divided into several sectors in the same slot on the same channel.

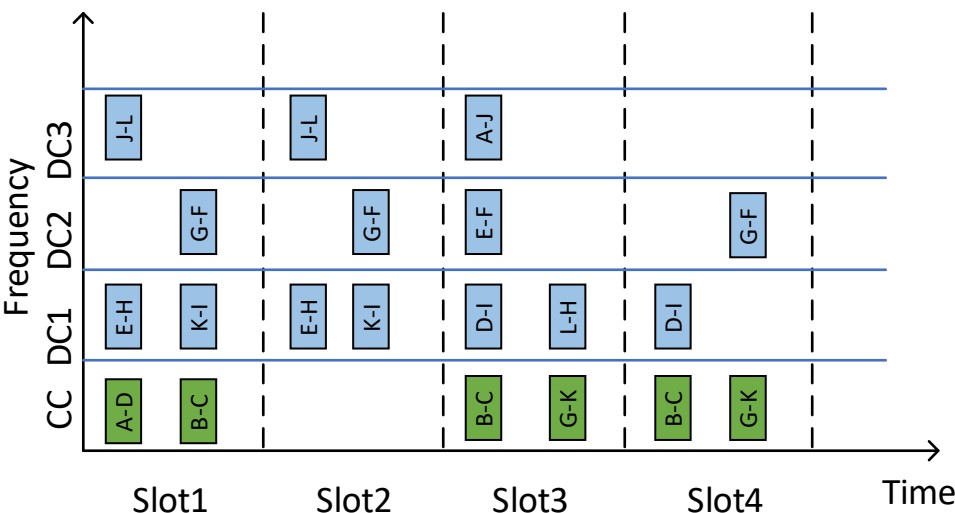

**Figure 3.** Directional multichannel network structure.

As the example illustrates in Figure 3, there are three data channels highlighted in bule, i.e., DC1, DC2, and DC3, along with a control channel highlighted in green, i.e., CC. A-D presents the directional communication between UAV A and UAV D. In slot1, there are six pairs of UAVs engaging in communication. Although A-D and B-C are in the same channel, there is no interference between A-D and B-C owing to the spatial reuse of directional communication, as shown in Figure 4.

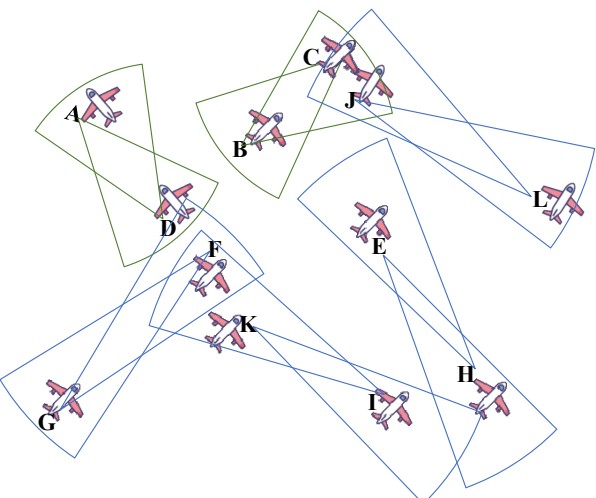

**Figure 4.** Directional multichannel communication in Slot1.

*2.3. Problem Definition*

The main objective is to minimize the worst-case-delay-to-main-lobe-rendezvous (WCDMR) without coordination or information for TS1. We consider a pair of neighbor UAVs, denoted as *a* and *b*. We assume that UAV *a* is located in the sector $P_b \in u$, and UAV *b* is located in the sector $P_a \in u$. UAV *a* points towards $I_a \in u$, and UAV *b* points towards $I_b \in u$ in initial state. The pair of neighboring UAVs, *a* and *b*, can only discover each other if and only if they steer their main lobes towards each other as illustrated in the following scenario.

In Figure 5, $N = 6$, $u = \{1, 2, 3, 4, 5, 6\}$, $P_a = 1$, $P_b = 4$, $I_a = 4$, and $I_b = 3$.

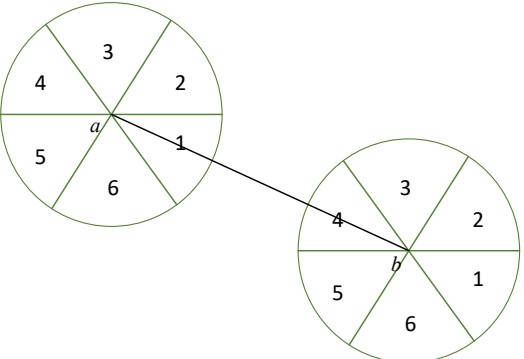

**Figure 5.** The communication of a pair of UAVs, *a* and *b*.

UAV *a* and UAV *b* switch their main beam to the next sector in one slot counterclockwise. The antenna scan sequence, denoted as **S**, is designed by different algorithms. The scan sequences of UAV *a* and UAV *b* are $S_a = \{4, 5, 6, 1, 2, 3\}$ and $S_b = \{3, 4, 5, 6, 1, 2\}$, respectively. The antenna scan sequences and the discovery process are illustrated in Figure 6. In Figure 6, the main lobes of the two UAVs cannot simultaneously point towards sector 1 and sector 4, so the pair of the UAVs cannot discover each other through this switching method.

| Slot index | 1 | 2 | 3 | 4 | 5 | 6 | 7 | 8 | 9 | 10 | 11 | 12 |
|---|---|---|---|---|---|---|---|---|---|---|---|---|
| UAV *a* | 4 | 5 | 6 | 1 | 2 | 3 | 4 | 5 | 6 | 1 | 2 | 3 |
| UAV *b* | 3 | 4 | 5 | 6 | 1 | 2 | 3 | 4 | 5 | 6 | 1 | 2 |

**Figure 6.** Antenna scan sequences of the above scenario.

For probabilistic neighbor discovery strategies, the period of **S** may be infinite; therefore, neighbor discovery delay cannot be guaranteed. Hence, the deterministic method is proposed.

Overall, the criteria for the deterministic method of main lobe blind rendezvous are shown as follows:

i.   Guaranteed rendezvous: any two UAVs can achieve main lobe rendezvous in a certain period of time.
ii.  Full rendezvous diversity: UAVs can rendezvous on any combination of $(P_a, P_b) \in [1, N] \times [1, N]$ and $(I_a, I_b) \in [1, N] \times [1, N]$.
iii. Asynchronous environment: in FANETs, it is difficult to employ highly tight time synchronization among users, and each user may start their scan sequence at a different time during initial access.
iv.  Without extra expenditure: no additional expenditures, such as a coordination channel and prior information, should be required.

The blind rendezvous problem for neighbor discovery with directional antennas is defined as follows:

$$
\begin{aligned}
&\min T_{\mathrm{br}} \\
&s.t. \forall P_a \in [1, N], P_b \in [1, N], \\
&\forall I_a \in [1, N], I_b \in [1, N] \\
&\exists t \le T_{\mathrm{br}}, S_{\mathrm{a}}(t) = P_a, S_{\mathrm{b}}(t) = P_a
\end{aligned}
\tag{3}
$$

To clarify, the blind rendezvous problem with directional antennas for neighbor discovery is used to design antenna scan sequences **S** that minimize the worst-case discovery delay $T_{\mathrm{br}}$, while guaranteeing neighbor discovery between any pair of neighbor UAVs $a$ and $b$ for any $(P_a, P_b)$ and $(I_a, I_b)$.

## 3. Multichannel MAC Protocol with Directional Antenna for FANET

In this section, we propose blind rendezvous algorithms with directional antenna for main lobe rendezvous of a pair of UAVs and a FANET under the circumstance of being without coordination or prior information, which are denoted as the BR-DA and BR-DA-FANET algorithm, respectively. Furthermore, the ND-LP protocol and ACI-LP protocol are proposed to further decrease delay.

### 3.1. Blind Rendezvous Algorithm with Directional Antenna

The basic idea of the BR-DA algorithm is inspired by the operation of a circular clock with two hands moving at different walking speeds. The hands of the clock walk clockwise at their respective speeds and will surely meet each other at a specific time. Similarly, in a FANET, any pairwise main lobe of UAVs can be assimilated to these two hands in the clock, and the different time durations of the main lobe in each sector can be regarded as the different walking speeds of these two hands. Thus, the main lobe of UAVs can achieve rendezvous.

We specify a time duration of $M_T T$ and $M_R T$ for the transmitter UAV and receiver UAV, respectively. $M_{\mathrm{T}}$ and $M_R$ are different, coprime, and positive integers. T is the minimal slot duration to exchange discovery beacons between the pair of UAVs.

The rendezvous period of the system composed of two UAVs is defined as the duration from the initial state to the last state. Note that during each period, the two UAVs search for all main lobe rendezvous cases according to the algorithms. Figure 7 illustrates the different state of the scenario of Section 2.3, and the arrows indicate the direction of the main lobe. It is important to note that the main lobe rendezvous state may not be achieved because full rendezvous cannot always be accomplished by different algorithms.

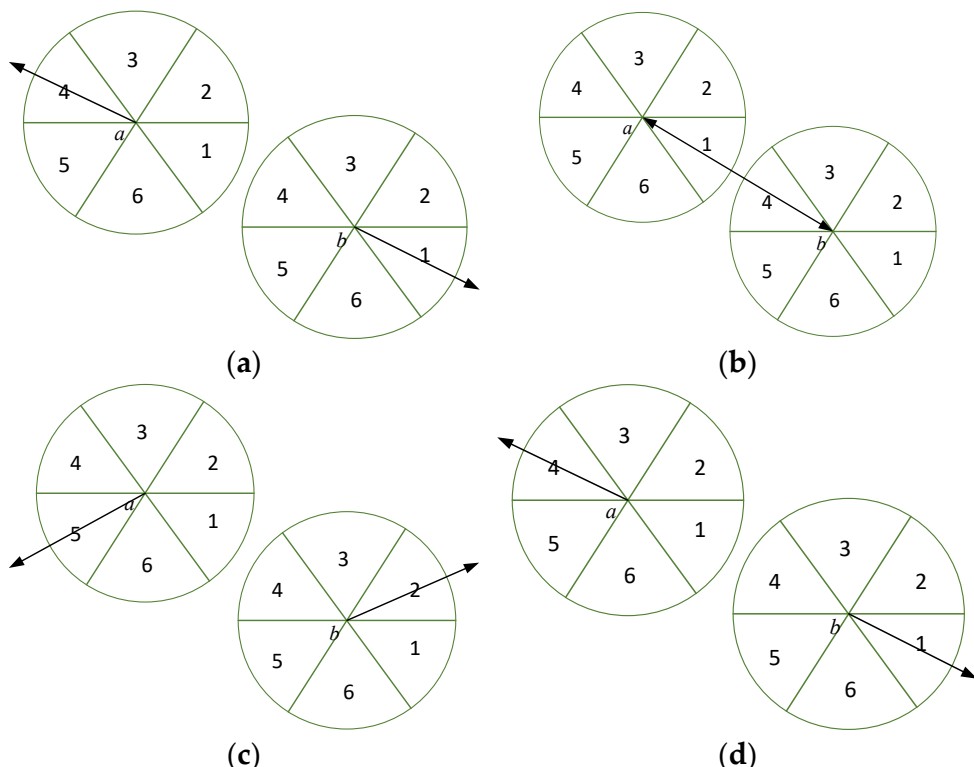

**Figure 7.** The rendezvous period of the system composed of two UAVs. (**a**) The initial state; (**b**) the main lobe rendezvous state; (**c**) the last state; and (**d**) the initial state of the next period.

**Theorem 1.** *As two UAVs steer their antennas towards each other, the rendezvous period of the system is* $M_T M_R N\text{T}$.

**Proof of Theorem 1.** The period of transmitter UAV and receiver UAV are $M_T N\text{T}$ and $M_R N\text{T}$ for scanning all directions, respectively. During each rendezvous period, the transmitter UAV and receiver UAV scan all directions and beamform towards the initial sectors they scan together. Hence, the rendezvous period of the system is the least common multiple of $M_T N\text{T}$ and $M_R N\text{T}$, i.e., $M_T M_R N\text{T}$. □

**Theorem 2.** $M_T$ *and* $M_R$ *need to satisfy the constraints* $(M_T = 1, M_R = N)$ *and* $(M_T = N, M_R = 1)$ *to achieve the theoretical supremum for clock synchronization discovery.* $(M_T = 1, M_R = N)$ *presents* $M_T = 1$ *and* $M_R = N$ *at the same time. The least delay of the full rendezvous algorithm is* $N^2\text{T}$ *if and only if the above constraint is satisfied for clock synchronization discovery. Similarly, the least-worst-case (supremum) delay of blind rendezvous with clock synchronization is* $N^2\text{T}$.

**Proof of Theorem 2.** The number of blind rendezvous cases for a pair of UAVs is $N^2$. When $N$ equals four, the number of blind rendezvous cases is 16, i.e., {(1,1), (1,2), (1,3), (1,4), (2,1), (2,2), (2,3), (2,4), (3,1), (3,2), (3,3), (3,4), (4,1), (4,2), (4,3), (4,4)}. Here, $(\cdot,\cdot)$ represents the sectors towards which the main lobe of a pair of UAVs beamform at the same slot. There is only one case in which they switch their antennas towards each other. If this pair of UAVs obtains full rendezvous diversity in a rendezvous period, the number of cases for this pair of UAVs is less than the rendezvous period, i.e., $M_T M_R N\text{T} \geq N^2\text{T}$. Therefore, we can get $M_T M_R \geq N$. In order to minimize the delay of the blind rendezvous algorithm, the rendezvous period of this pair of UAVs equals the number of cases, i.e., $M_T M_R = N$. □

When the constraints $(M_T \neq 1, M_R \neq N)$ and $(M_T \neq N, M_R \neq 1)$ are applied to $M_T M_R = N$, a full rendezvous diversity cannot be achieved. Moreover, under the constraints $(M_T \neq 1, M_R \neq N)$ and $(M_T \neq N, M_R \neq 1)$, we can obtain $M_T \geq 2$ and $M_R \geq 2$. In two continuous durations T, the sectors that the two UAVs beamform towards do not switch. In a rendezvous period, there are $M_T M_R N - \frac{M_T M_R N}{\min(M_T, M_R)}$ cases that cannot be searched. In order to scan all cases, it takes $M_T M_R NT - \max(M_T, M_R)NT$ to scan for the system composed of this pair of UAVs. Only under the constraints $(M_T = 1, M_R = N)$ and $(M_T = N, M_R = 1)$ can the minimum delay of full rendezvous diversity, i.e., $M_T M_R NT$, be achieved. Consequently, the least WCDMR without coordination or information is $N^2 T$ for clock synchronization discovery.

In the scenario described in Section 2.3, if $(M_T = 2, M_R = 3)$, the antenna scan sequences for UAVs $a$ and $b$ are shown in Figure 8. UAV $a$ switches to sector 1, but UAV $b$ cannot switch to sector 4 at the same slot from Figure 8.

| Slot index | 1 | 2 | 3 | 4 | 5 | 6 | 6 | 7 | 8 | 9 | 10 | 11 | 12 | 13 | 15 | 16 | 17 | 18 |
|---|---|---|---|---|---|---|---|---|---|---|---|---|---|---|---|---|---|---|
| UAV $a$ | 4 | 4 | 5 | 5 | 6 | 6 | 1 | 1 | 2 | 2 | 3 | 3 | 4 | 4 | 5 | 5 | 6 | 6 |
| UAV $b$ | 3 | 3 | 3 | 4 | 4 | 4 | 5 | 5 | 5 | 6 | 6 | 6 | 1 | 1 | 1 | 2 | 2 | 2 |

| | 19 | 20 | 21 | 22 | 23 | 24 | 25 | 26 | 27 | 28 | 29 | 30 | 31 | 32 | 33 | 34 | 35 | 36 |
|---|---|---|---|---|---|---|---|---|---|---|---|---|---|---|---|---|---|---|
| UAV $a$ | 1 | 1 | 2 | 2 | 3 | 3 | 4 | 4 | 5 | 5 | 6 | 6 | 1 | 1 | 2 | 2 | 3 | 3 |
| UAV $b$ | 3 | 3 | 3 | 4 | 4 | 4 | 5 | 5 | 5 | 6 | 6 | 6 | 1 | 1 | 1 | 2 | 2 | 2 |

**Figure 8.** Antenna scan sequences for UAV $a$ and $b$ if $(M_T = 2, M_R = 3)$.

If $(M_T = 1, M_R = 6)$, the antenna scan sequences for UAV $a$ and $b$ are shown in Figure 9. In slot 9, UAV $a$ switches to sector 1, and UAV $b$ can switch to sector 4 at the same slot. In the scenario of Section 2.3, there are 36 cases of the scenario, and it takes at least 36 slots to achieve full rendezvous of the main lobe. In the 36 slots, if $(M_T = 1, M_R = 6)$, the full rendezvous diversity is achieved.

| Slot index | 1 | 2 | 3 | 4 | 5 | 6 | 6 | 7 | 8 | 9 | 10 | 11 | 12 | 13 | 15 | 16 | 17 | 18 |
|---|---|---|---|---|---|---|---|---|---|---|---|---|---|---|---|---|---|---|
| UAV $a$ | 4 | 5 | 6 | 1 | 2 | 3 | 4 | 5 | 6 | 1 | 2 | 3 | 4 | 5 | 6 | 1 | 2 | 3 |
| UAV $b$ | 3 | 3 | 3 | 3 | 3 | 3 | 4 | 4 | 4 | 4 | 4 | 4 | 5 | 5 | 5 | 5 | 5 | 5 |

| | 19 | 20 | 21 | 22 | 23 | 24 | 25 | 26 | 27 | 28 | 29 | 30 | 31 | 32 | 33 | 34 | 35 | 36 |
|---|---|---|---|---|---|---|---|---|---|---|---|---|---|---|---|---|---|---|
| UAV $a$ | 4 | 5 | 6 | 1 | 2 | 3 | 4 | 5 | 6 | 1 | 2 | 3 | 4 | 5 | 6 | 1 | 2 | 3 |
| UAV $b$ | 6 | 6 | 6 | 6 | 6 | 6 | 1 | 1 | 1 | 1 | 1 | 1 | 2 | 2 | 2 | 2 | 2 | 2 |

**Figure 9.** Antenna scan sequences for UAV $a$ and $b$ if $(M_T = 1, M_R = 6)$.

For achieving a minimum worst-case delay of blind rendezvous, Theorem 2 can be employed for clock synchronization discovery but not for clock asynchronization discovery. There are several blind rendezvous cases in which the time duration is less than T, so blind rendezvous with directional antennas cannot be achieved in $N^2 T$. In Figure 10, according to Theorem 2, the time duration of the cases (1,4) and (3,6) equals T and can be searched for clock synchronization discovery. In Figure 11, according to Theorem 2, the time duration of the case (1,4) is more than T, enabling the neighbor discovery of this case for clock asynchronization discovery. However, the time duration of the case (3,6) is less than T, making it impossible to accomplish neighbor discovery for clock asynchronization discovery. From the above analysis, full rendezvous diversity cannot be attained for clock asynchronization discovery according to Theorem 2.

| Slot index | 1 | 2 | 3 | 4 | 5 | 6 | 6 | 7 | 8 | 9 | 10 | 11 | 12 | 13 | 15 | 16 | 17 | 18 |
|---|---|---|---|---|---|---|---|---|---|---|---|---|---|---|---|---|---|---|
| UAV *a* | 4 | 5 | 6 | 1 | 2 | 3 | 4 | 5 | 6 | 1 | 2 | 3 | 4 | 5 | 6 | 1 | 2 | 3 |
| UAV *b* | 3 | 3 | 3 | 3 | 3 | 3 | 4 | 4 | 4 | 4 | 4 | 4 | 5 | 5 | 5 | 5 | 5 | 5 |

| | 19 | 20 | 21 | 22 | 23 | 24 | 25 | 26 | 27 | 28 | 29 | 30 | 31 | 32 | 33 | 34 | 35 | 36 |
|---|---|---|---|---|---|---|---|---|---|---|---|---|---|---|---|---|---|---|
| UAV *a* | 4 | 5 | 6 | 1 | 2 | 3 | 4 | 5 | 6 | 1 | 2 | 3 | 4 | 5 | 6 | 1 | 2 | 3 |
| UAV *b* | 6 | 6 | 6 | 6 | 6 | 6 | 1 | 1 | 1 | 1 | 1 | 1 | 2 | 2 | 2 | 2 | 2 | 2 |

**Figure 10.** Antenna scan sequences for clock synchronization discovery of UAV *a* and *b* according to Theorem 2.

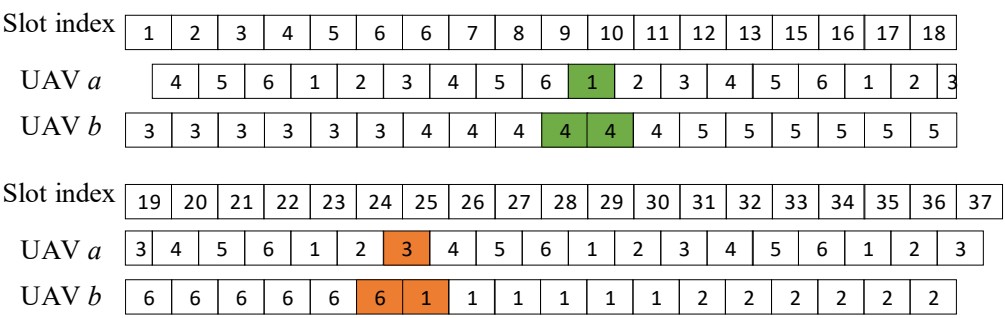

| Slot index | 1 | 2 | 3 | 4 | 5 | 6 | 6 | 7 | 8 | 9 | 10 | 11 | 12 | 13 | 15 | 16 | 17 | 18 |
|---|---|---|---|---|---|---|---|---|---|---|---|---|---|---|---|---|---|---|
| UAV *a* | 4 | 5 | 6 | 1 | 2 | 3 | 4 | 5 | 6 | 1 | 2 | 3 | 4 | 5 | 6 | 1 | 2 | 3 |
| UAV *b* | 3 | 3 | 3 | 3 | 3 | 3 | 4 | 4 | 4 | 4 | 4 | 4 | 5 | 5 | 5 | 5 | 5 | 5 |

| Slot index | 19 | 20 | 21 | 22 | 23 | 24 | 25 | 26 | 27 | 28 | 29 | 30 | 31 | 32 | 33 | 34 | 35 | 36 | 37 |
|---|---|---|---|---|---|---|---|---|---|---|---|---|---|---|---|---|---|---|---|
| UAV *a* | 3 | 4 | 5 | 6 | 1 | 2 | 3 | 4 | 5 | 6 | 1 | 2 | 3 | 4 | 5 | 6 | 1 | 2 | 3 |
| UAV *b* | 6 | 6 | 6 | 6 | 6 | 6 | 1 | 1 | 1 | 1 | 1 | 1 | 2 | 2 | 2 | 2 | 2 | 2 | |

**Figure 11.** Antenna scan sequences for clock asynchronization discovery of UAV *a* and *b* according to Theorem 2.

**Theorem 3.** *The minimum delay of the full rendezvous algorithm is achieved only when the constraints $(M_T = 1, M_R = N + 1)$ and $(M_T = N + 1, M_R = 1)$ are satisfied for clock asynchronization discovery. Additionally, the least-worst-case delay (supremum) of blind rendezvous without clock synchronization is $N(N + 1)T$.*

**Proof of Theorem 3.** With the constraints $(M_T = 1, M_R = N + 1)$ and $(M_T = N + 1, M_R = 1)$, the time duration of all cases is more than T; hence, the full rendezvous diversity can be achieved for clock asynchronization discovery. Under the constraints $(M_T = 1, M_R = N + 1)$ and $(M_T = N + 1, M_R = 1)$, the delay of blind rendezvous is $N(N + 1)T$. $\square$

Based on the above analysis, we can conclude that under the constraints $(M_T = 1, M_R = N)$ and $(M_T = N, M_R = 1)$, the full rendezvous diversity cannot be achieved, and hence, we can get that the minimum $M_T M_R$ is $N + 1$ for asynchronous neighbor discovery. Under the constraints $(M_T \neq 1, M_R \neq N + 1)$, $(M_T \neq N + 1, M_R \neq 1)$, and $M_T M_R = N + 1$, we can get and $M_R \geq 2$. In $N(N + 1)T$, there are at least $N^2 - \max(M_T, M_R)N$ cases that cannot be searched. Therefore, under the constraints $(M_T = 1, M_R = N + 1)$ and $(M_T = N + 1, M_R = 1)$, the least delay of full rendezvous can be achieved for clock asynchronization discovery and is $N(N + 1)T$. Under the constraints $(M_T = 1, M_R = N + 1)$ and $(M_T = N + 1, M_R = 1)$, the least WCDMR without coordination or information is $N(N + 1)T$ for clock asynchronization discovery.

In Figure 12, according to Theorem 3, the time duration of the cases (1,4) and (3,6) is greater than T, and thus, the neighbor discovery of the two cases can be accomplished for clock asynchronization discovery.

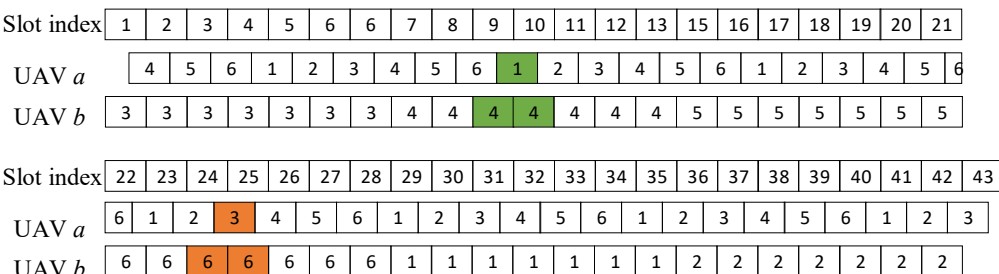

**Figure 12.** Antenna scan sequences for clock asynchronization discovery of UAV $a$ and $b$ according to Theorem 3.

The least delay and the worst-case delay of Theorem 2 for clock synchronization discovery are T and $N^2$T, respectively. The average delay of Theorem 2 for clock synchronization discovery is expressed as follows, where Z represents a positive integer:

$$T_{Sy-Bave} = \frac{1}{N^2} \sum_{\substack{i \in [0, N-1] \cup Z \\ j \in [1, N] \cup Z}} (iN + j)T = \frac{1 + N^2}{2} T \tag{4}$$

The least delay and the worst-case delay of Theorem 3 for clock asynchronization discovery are 2T and $N(N+1)$T, respectively. The average delay of Theorem 3 for clock asynchronization discovery is expressed as follows:

$$
\begin{aligned}
T_{ANSy-Bave} &= \frac{1}{N^2} \sum_{\substack{i \in [0, N-1] \cup Z \\ j \in [2, N+1] \cup Z}} (iN + j)T \\
&= \frac{N^2 + N + 2}{2} T
\end{aligned}
\tag{5}
$$

*3.2. Main Lobe Rendezvous Scheme-Based Blind Rendezvous Algorithm with Directional Antenna for FANET*

In Section 3.1, the BR-DA algorithm is used for the main lobe rendezvous of two UAVs. In this section, we introduce the BR-DA-FANET algorithm for the main lobe rendezvous of the FANET and the discovery beacon exchange scheme.

In a FANET, each UAV cannot receive the mode of other UAVs. For neighbor discovery of a FANET, the main lobe of any two UAVs should be rendezvoused. Moreover, one UAV should be in transmission mode, and the other should be in reception mode simultaneously. A specific control sequence with a length of $L$ bits is used to ensure the different modes for two neighboring UAVs in a FANET, as described in [15,24].

We define a rendezvous-slot as an interval with duration $T_{rs}$. $T_{rs}$ equals $N^2$T and $2N(N+1)$T for clock synchronization discovery and clock asynchronization discovery, respectively. That is, in each rendezvous-slot, each UAV in a FANET can achieve blind rendezvous. If a rendezvous-slot is marked with a "1", the corresponding UAV operates in the transmission mode; otherwise, it operates in the reception mode. We can assign a specific sequence of 0′s and 1′s to each UAV to control the UAV's operation mode during the neighbor discovery process without centralized control.

According to [15,24], the specific control sequence consists of an $l_1$-bit segment of 1′s, followed by the UAV's unique ID whose length is $l_2$ bits and has an $l_0$-bit segment of 0′s. The relationship of $l_0$, $l_1$, and $l_2$ is expressed as:

$$
\begin{cases}
L = 2l_2 + 1 \\
l_2 + 1 = l_0 + l_1
\end{cases}
\tag{6}
$$

In [15,24], the authors have proved that any two different specific control sequences can guarantee that the two UAVs can operate in different modes for at least one-bit duration

considering all possible cyclic rotations within a sequence of $L$ consecutive bits. The BR-DA-FANET algorithm achieves main beam rendezvous for the FANET through the combination of specific control sequences and BR-DA.

The least-worst-case discovery delay based on the BR-DA-FANET algorithm for clock synchronization discovery is expressed as (7).

$$T_{Sy-BNetMAX}= T_{rs} \times L = LN^2T \tag{7}$$

The maximum number of UAVs in a FANET is $U$. Therefore, the least-worst-case discovery delay of the BR-DA-FANET algorithm for clock synchronization discovery is expressed as (8), where $\lceil \cdot \rceil$ represents the ceiling function.

$$T_{Sy-BNetMAX}= T_{rs} \times L = LN^2T= (2\left\lceil \log_2^U \right\rceil + 1)N^2T \tag{8}$$

**Theorem 4.** *$2T_{rs}$ is the least duration for a rendezvous-slot to achieve main lobe blind rendezvous for a FANET. The least-worst-case discovery delay based on the BR-DA-FANET for clock asynchronization discovery of a FANET is $2LN(N+1)T$.*

**Proof of Theorem 4.** There is a drift ranging from 0 to one $T_{rs}$ for the clock asynchronization discovery of a FANET. $\square$

In Figure 13, Figure 13b represents Figure 13a with a drift ranging from 0 to 0.5 rendezvous-slot. When the drift ranges from 0 to 0.5 rendezvous-slot, the aligned time of any two UAVs' rendezvous-slot falls within the interval of $0.5T_{rs}$ to $T_{rs}$. In order to achieve full rendezvous of the main lobe for a FANET, the aligned time of any two UAVs in the FANET must be greater than $N(N+1)T$. We define a rendezvous-slot as an interval with duration $2N(N+1)T$. Therefore, the main lobe rendezvous of a FANET can be achieved with drift which ranges from 0 to 0.5 rendezvous-slot.

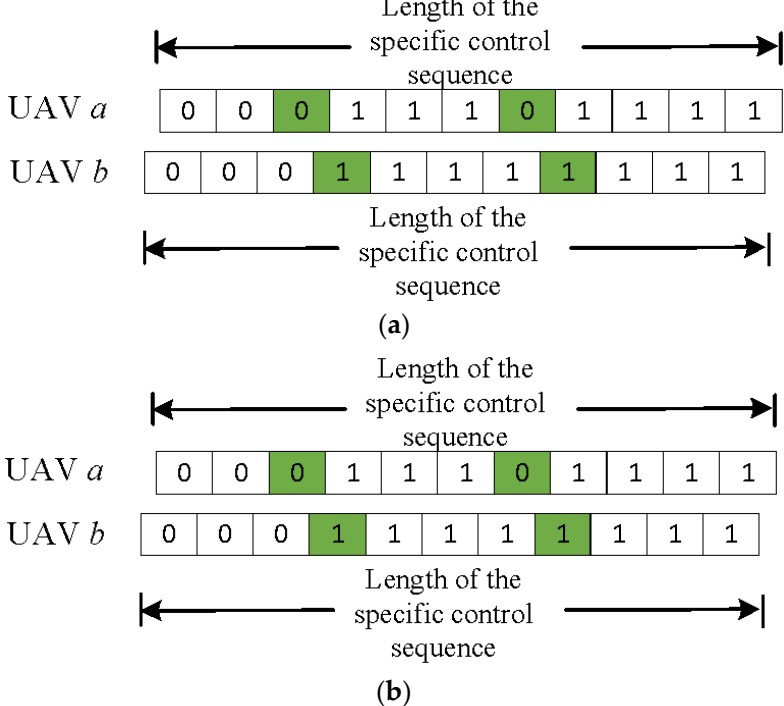

**Figure 13.** The picture for drift which ranges from 0 to $0.5T_{rs}$. (**a**) The rendezvous-slots of the two UAVs are aligned for clock synchronization discovery of a FANET; (**b**) drift ranges from 0 to 0.5 rendezvous-slot for clock asynchronization discovery of a FANET.

Figure 14a represents Figure 13a with drift ranging from 0.5 rendezvous-slot to 1 rendezvous-slot. When the drift ranges from 0.5 rendezvous-slot to 1 rendezvous-slot, the aligned time of any two UAVs' rendezvous-slot falls within the interval of 0 to 0.5 rendezvous-slot. In order to achieve main lobe rendezvous for a FANET, the aligned time of any two UAVs in the FANET must be greater than $N(N+1)T$. As the aligned time approaches 0, the rendezvous-slot must approach $+\infty$, so the delay of neighbor discovery is $+\infty$.

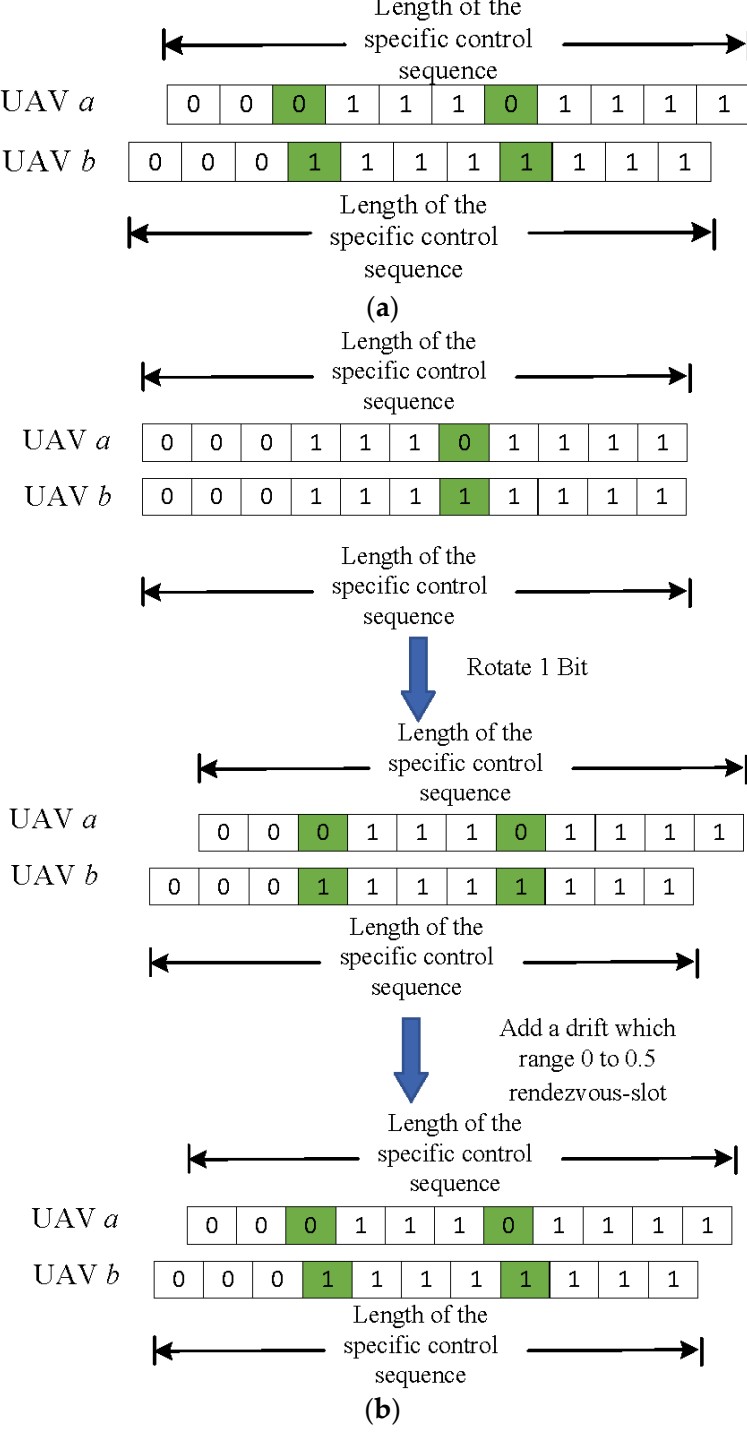

**Figure 14.** The picture for drift which ranges from 0.5 rendezvous-slot to 1 rendezvous-slot. (**a**) Drift ranges from 0.5 rendezvous-slot to 1 rendezvous-slot for clock asynchronization discovery of a FANET. (**b**) The process to add a drift which ranges from 0.5 rendezvous-slot to 1 rendezvous-slot.

In Figure 13a, we introduce a rotation of 1 bit to one of the two specific control sequences, as shown in Figure 14b. According to [15,24], the rotated sequences still guarantee that the two UAVs can operate in different modes for a duration of at least one bit. We then apply a drift ranging from 0 to 0.5 rendezvous-slot to the sequence that underwent the rotation, as displayed in Figure 14b. Through this process, the two specific control sequences achieve the addition of a drift ranging from 0.5 rendezvous-slot to 1 rendezvous-slot, as illustrated in Figure 14a. We define a rendezvous-slot as an interval with duration $2N(N+1)T$. The main lobe rendezvous of a FANET can be achieved with drift which ranges from 0.5 rendezvous-slot to 1 rendezvous-slot.

Based on the aforementioned analysis, we define a rendezvous-slot as an interval with duration $2N(N+1)T$, and thus the main lobe rendezvous of a FANET can be achieved for clock asynchronization discovery. $2N(N+1)T$ is the smallest duration for a rendezvous-slot to achieve main lobe blind rendezvous for a FANET.

The least-worst-case discovery delay based on the BR-DA-FANET algorithm for clock asynchronization discovery is expressed as (9).

$$T_{ANSy-BNet\,supremum} = T_{rs} \times L = 2LN(N+1)T \tag{9}$$

As the maximum number of UAVs in a FANET is $U$, the least-worst-case discovery delay of the BR-DA-FANET algorithm for clock asynchronization discovery is expressed as (10).

$$T_{ANSy-BNet\,supremum} = T_{rs} \times L = 2LN(N+1)T$$
$$= 2(2\left\lceil \log_2^U \right\rceil + 1)N(N+1)T \tag{10}$$

After achieving main lobe rendezvous, the discovery beacon schedule is utilized for neighbor discovery. The discovery beacon schedule is used for neighbor discovery in T. In this paper, the discovery beacon schedule introduced in [24] is employed. The BR-DA and BR-DA-FANET algorithms combined with the discovery beacon schedule are the BR-DA and BR-DA-FANET protocols, respectively.

BARTS and BACTS, which are shown in Figure 15, are used for neighbor discovery. "Type" indicates whether the frame is a control frame or a data frame. "Subtype" signifies whether this frame belongs to the subtype BARTS or BACTS. "ID" indicates the ID sequence of the transmitting UAV. "Location" represents the location of the transmitting UAV from the global positioning system (GPS) at the transmission time. "Time" provides information on clock synchronization and the transmission time. Additionally, "Speed" and "Direction" indicate the speed and the movement direction of the transmitting UAV, respectively.

| Type | Subtype | ID | Location | Time | Speed | Direction |
|------|---------|-----|----------|------|-------|-----------|

**Figure 15.** Frame structure of BARTS and BACTS.

From the received BARTS and BACTS, the receiving UAV can receive the ID sequence, location, time, speed, and direction of the transmitting UAV.

### 3.3. Neighbor Discovery with Location Prediction for FANET

According to the BR-DA-FANET protocol, UAVs in a FANET can discover neighbors without coordination or prior information. Once the initial access is established, each UAV in the FANET obtains essential information about its neighbors, including their ID, location, speed, direction, and clock synchronization. According to this information, UAVs can reduce the neighbor discovery delay.

In this section, we propose the neighbor discovery with location prediction (ND-LP) protocol for a FANET from this information to achieve quick main lobe rendezvous and channel rendezvous simultaneously. The sector of the switched beam antenna model is big,

and the time between two neighbor discoveries is short. Hence, we can consider that UAVs move in a specific direction with a constant velocity.

The location and speed of the *i*-th UAV in a FANET are marked as $(x_i, y_i)$ and $v_i$, respectively. The angle between the movement direction of the *i*-th UAV and the positive direction of the *x*-axis is $\zeta_i$. This information is transmitted at time $t_i$. At time $t_j$, UAV *j* transmits information to UAV *i*, and the location of UAV *i*, which is predicted by UAV *j*, is shown as:

$$\begin{cases} x_i' = x_i + v_i(t_j - t_i)\cos\zeta_i \\ y_i' = y_i + v_i(t_j - t_i)\sin\zeta_i \end{cases} \tag{11}$$

According to (11), UAV *a*, which prepares to transmit information to neighboring UAV *b*, predicts the location of UAV *b* in the FANET. Then, the main lobe of UAV *a* points toward UAV *b*, and UAV *a* transmits TS1-RTS, as illustrated in Figure 16, to UAV *b*. When the main lobe UAV *b* points toward UAV *a*, UAV *b* can receive TS1-RTS. Subsequently, UAV *b* responds with TS1-CTS, as shown in Figure 16, to UAV *a*. According to the process, quick neighbor discovery can be achieved. "Receiver ID" and "transmitter ID" refer to the ID of the received UAV and transmitted UAV of this frame, respectively. "Channel" in TS1-RTS represents the available channels for transmitting data, while "Duration" indicates the transmission duration. "Channel" in TS1-CTS indicates the selected channel which is available for UAV *a* and UAV *b*.

| Type | Subtype | Duration | Receiver ID | Transmitter ID | Channel | Location | Time | Speed | Direction |
|------|---------|----------|-------------|----------------|---------|----------|------|-------|-----------|

**Figure 16.** The frame structure of TS1-RTS and TS1-CTS.

In this paper, we focus on the one-hop of TS2. When UAV *a* intends to transmit data to UAV *b*, it first judges whether UAV *b* is in the one-hop range of TS1 of UAV *a*. If UAV *b* is in the one-hop range of TS1 of UAV *a*, UAV *a* discovers UAV *b* according to the ND-LP protocol. UAV *b* chooses the available channel which is available for TS2 of UAV *a* and UAV *b* and transmits the available channel to TS1 of UAV *a* through TS1-CTS. UAV *b* predicts the location of UAV *a* through (11), and then, TS2 of UAV *b* points to UAV *a*. TS2 of UAV *b* switches to the chosen channel and reception mode. From location prediction-based TS1-CTS, TS2 of UAV *a* points to UAV *b* and switches to the chosen channel. Main lobe rendezvous and channel rendezvous are achieved at the same time. The described process above is pictured in Figure 17.

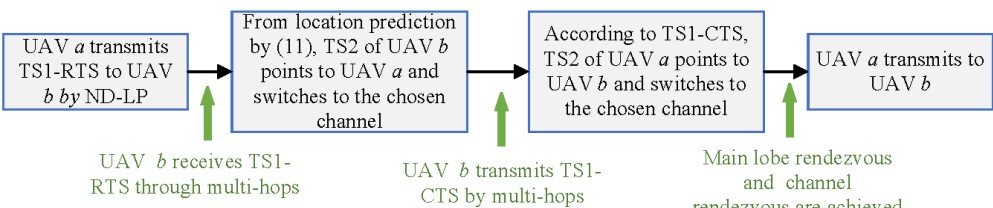

**Figure 17.** Main lobe rendezvous and channel rendezvous are achieved in one-hop distance of UAV *a* TS1 as well as UAV *b* TS1 according to ND-LP.

In order to discover all neighbors of TS2 according to TS1, the multihop method is used to extend the communication range for TS1.

If UAV *b* is beyond the one-hop range for TS1 of UAV *a*, TS1 of UAV *a* transmits TS1-RTS2 (as shown in Figure 18) to UAV *b* through multihop communication. Based on the location prediction obtained from TS1-RTS2, TS2 of UAV *a* points to UAV *b* and switches to the chosen channel. Subsequently, TS1 of UAV *b* responds with TS1-CTS2 (as shown in Figure 18) to UAV *a* according to the multihop method. Leveraging the location prediction, TS2 of UAV *a* points to UAV *b* and switches to the chosen channel. The process is shown in Figure 19.

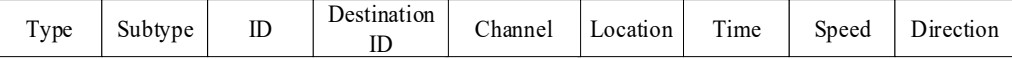

| Type | Subtype | ID | Destination ID | Channel | Location | Time | Speed | Direction |
| --- | --- | --- | --- | --- | --- | --- | --- | --- |

**Figure 18.** The frame structure of TS1-RTS2 and TS1-CTS2.

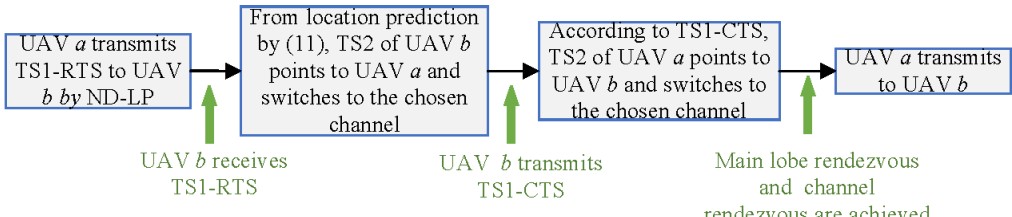

**Figure 19.** Main lobe rendezvous and channel rendezvous are achieved in a multihops distance of UAV *a* TS1 as well as UAV *b* TS1 according to ND-LP.

In Figure 18, "Destination ID" and "ID" refer to the ID of UAV *b* and the ID of UAV *a*, respectively. "Channel" in TS1-RTS2 indicates the available channels for TS2 of UAV *a*. "Channel" in TS1-CTS2 represents the chosen channel that is available for TS2 of UAV *a* and UAV *b*.

*3.4. Protocol for Avoiding Communication Interruption with Location Prediction*

In this section, we propose the avoiding communication interruption with location prediction (ACI-LP) protocol.

In a FANET, the continuous movement of UAVs can be susceptible to directional transmission interruptions. To address this issue, each data packet adds the duration of transmission for clock synchronization information, location, speed, and motion direction. Upon receiving the data packet, the receiver sends an ACK that contains clock synchronization information, location, speed, and direction information to the transmitter. Using the information from the data packet or ACK, the receiver can calculate the transmitter location through (11) and distance between two UAVs through (12). Subsequently, the receiver can judge whether the distance is within the communication distance of TS2. In (12), Δt denotes the duration of the transmitted packet, while $(x_\text{T}, y_\text{T})$ and $(x_\text{R}, y_\text{R})$ represent the transmitter location and receiver location, respectively.

$$D_\text{T,R} = \sqrt{\begin{array}{l}|x_\text{T} - x_\text{R} + v_\text{T}\Delta t \cos \zeta_\text{T} - v_\text{R}\Delta t \cos \zeta_\text{R}|^2 \\ + |y_\text{T} - y_\text{R} + v_\text{T}\Delta t \sin \zeta_\text{T} - v_\text{R}\Delta t \sin \zeta_\text{R}|^2\end{array}} \leq D_\text{max} \qquad (12)$$

To avoid interruptions in main lobe rendezvous caused by the high mobility of UAVs, it is necessary to predict the communication angle. For instance, in Figure 20, UAV *A* receives the data packet from UAV *B* and prepares to respond with an ACK to UAV *B*. The angle between the straight line AB and the straight line A′B′ is calculated through (13). $D_\text{max}$ is the maximum communication distance. The straight line A′B′ is the straight line AB after motion and is predicted by (14) and (15). $(x_A, y_A)$ and $(x_B, y_B)$ represent the locations of UAV *A* and UAV *B* at the time of the last main lobe rendezvous, respectively. $(x'_A, y'_A)$ and $(x'_B, y'_B)$ represent the predicted locations of UAV *A* and UAV *B* as the ACK is received successfully by UAV *B*, respectively, which are calculated using (14) and (15). $T_{CS}$ is the delay for sensing and judging channels availability. $T_{TS2-RTS}$ and $T_{TS2-CTS}$ are the delays of TS2-RTS and TS2-CTS, respectively. $T_{\text{Inf}}$ represents the duration of the preparation-to-transmit information, which corresponds to the duration of the ACK in the given example. Δt is the sum of the duration of the data packet and $T_{\text{Inf}}$. $\zeta_{B-\text{tran}}$ represents the angle between the movement direction of UAV B and the positive x-axis direction at the time of transmitting the ACK. Through (13), if $\alpha \leq \frac{\theta_\text{P}}{2}$, the next communication is not interrupted, and UAV *A* transmits ACK to UAV *B*. If $\alpha > \frac{\theta_\text{P}}{2}$, UAV *A* transmits TS2-RTS

to UAV *B*, and UAV *B* responds with TS2-CTS to UAV *A* in order to achieve main lobe rendezvous again. The frame structure of TS2-RTS and TS2-CTS is shown in Figure 21.

$$\alpha = \cos^{-1} \frac{(x_A - x_B, y_A - y_B).(x'_A - x'_B, y'_A - y'_B)}{\left|(x_A - x_B, y_A - y_B)\right| * (x'_A - x'_B, y'_A - y'_B)} \le \frac{\pi}{N} \tag{13}$$

$$\begin{cases} x'_A = x_{A\text{-tran}} + v_{A\text{-tran}}(T_{CS} + T_{TS2-RTS} + T_{TS2-CTS} + T_{\text{Inf}}) \cos \zeta_A \\ y'_A = y_{A\text{-tran}} + v_{A\text{-tran}}(T_{CS} + T_{TS2-RTS} + T_{TS2-CTS} + T_{\text{Inf}}) \sin \zeta_A \end{cases} \tag{14}$$

$$\begin{cases} x'_B = x_{B\text{-tran}} + v_{B\text{-tran}}(T_{CS} + T_{TS2-RTS} + T_{TS2-CTS} + \Delta t) \cos \zeta_{B\text{-tran}} \\ y'_B = y_{B\text{-tran}} + v_{B\text{-tran}}(T_{CS} + T_{TS2-RTS} + T_{TS2-CTS} + \Delta t) \sin \zeta_{B\text{-tran}} \end{cases} \tag{15}$$

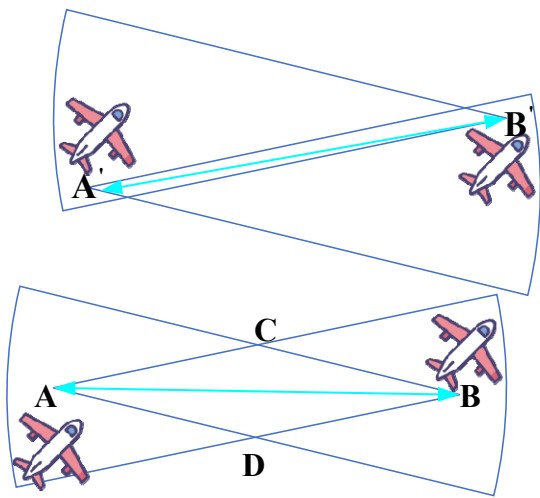

**Figure 20.** Main lobe rendezvous before transmitting ACK and after transmitting ACK.

| Type | Subtype | Duration | Receiver ID | Transmitter ID | Channel | Location | Time | Speed | Direction |
|---|---|---|---|---|---|---|---|---|---|
| | | | | | | | | | |

**Figure 21.** The frame structure of TS2-RTS and TS2-CTS.

The aforementioned process is depicted in Figure 22. The data transmission follows the same procedure as the ACK process.

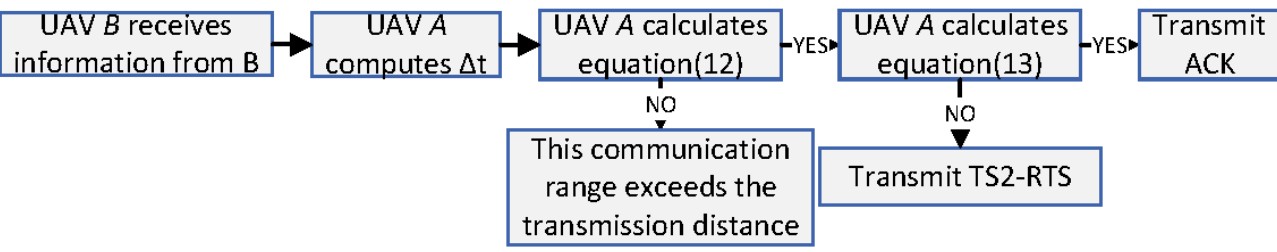

**Figure 22.** The process for avoiding communication interruption.

## 4. Simulation and Analysis

The ODND in [24] and the HDND in [15] are the latest directional neighbor discovery protocols without coordination or prior information that do not differentiate channel environments. The ODND protocol shares the same switched beam antenna model as the one we adopt, making the comparison with the ODND protocol more equitable. However, the continuous scanning directional antenna model in the HDND protocol is relatively ideal and, in practice, is challenging to implement in engineering due to beam granularity [35].

### 4.1. Simulation on Pair-Wise Neighbor Discovery

We implemented the BR-DA protocol to validate the neighbor discovery process between a pair of UAVs. We traced the discovery delay for the different number of sectors and compared it with the ODND protocol presented in [24]. $P_a$, $P_b$, $I_a$, and $I_b$ were randomly generated. The clock drift between UAV $a$ and UAV $b$ was randomly generated from [0, 1000] slots. Both UAV $a$ and UAV $b$ were assigned 8-bit random IDs. The worst-case discovery delay and average discovery delay are the maximal delay and average discovery delay of 10,000 runs, respectively. Additionally, we computed the duration of 10,000 runs and compared it with the ODND protocol.

The simulation results, based on 10,000 runs, demonstrate that the BR-DA protocol exhibits lower calculation complexity compared to the ODND protocol. In Figure 23, the duration of 10,000 runs for BR-DA with different numbers of sectors is consistently lower than that of the ODND protocol.

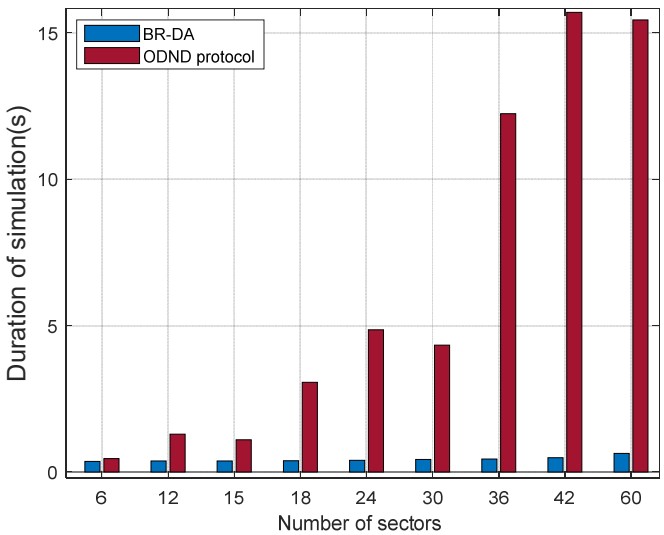

**Figure 23.** The duration of 10,000 runs for different numbers of sectors.

The worst-case discovery delay for different numbers of sectors is shown in Figure 24, clearly showing that the BR-DA protocol achieves a lower worst-case discovery delay compared to the ODND protocol.

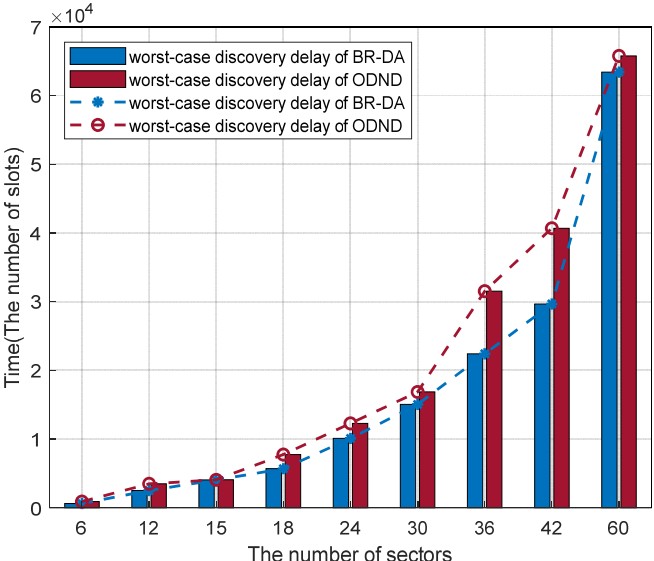

**Figure 24.** The worst-case discovery delay for different numbers of sectors.

Figure 25 illustrates the average discovery delay for different numbers of sectors, indicating that as the number of sectors approaches $2^n (2^{n-1} < N \leq 2^n, n > 1, n \in Z)$, the average discovery delay for the BR-DA protocol becomes significantly lower than that of the ODND protocol. In order to validate this conclusion, we conducted further simulations for the average discovery delay by varying the number of sectors from 8 to 32, as depicted in Figure 26. The results indicate that in cases where the number of sectors ranges from 12 to 16 and from 24 to 32, the average delay of the BR-DA protocol is less than that of the ODND protocol. Consequently, we can make the conclusion that the average discovery delay of the BR-DA protocol is less than that of the ODND protocol as the number of sectors is close to $2^n (2^{n-1} < N \leq 2^n, n > 1, n \in Z)$. This result can be attributed to the diminishing randomness of the ODND protocol as the number of sectors approaches $2^n (2^{n-1} < N \leq 2^n, n > 1, n \in Z)$, as stated in [24]. The average neighbor discovery delay of the random neighbor discovery protocol is minimal [15], but there is no upper bound on the neighbor discovery delay. The ODND protocol achieves a lower neighbor discovery delay by increasing the protocol's randomness, but this will increase the worst-case discovery delay.

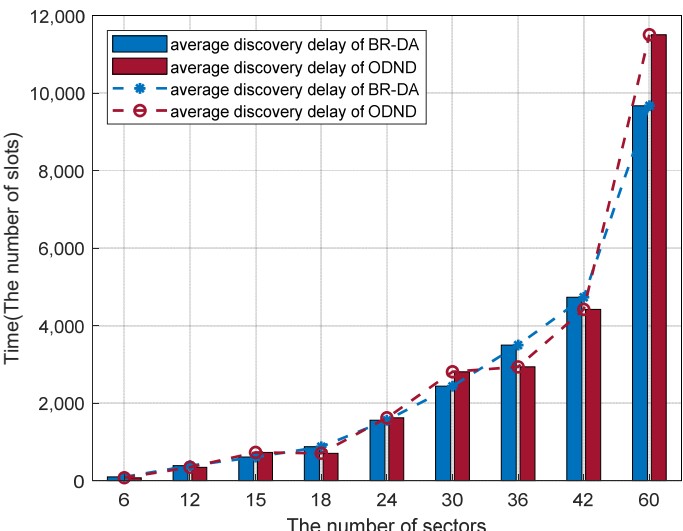

**Figure 25.** The average discovery delay for different numbers of sectors.

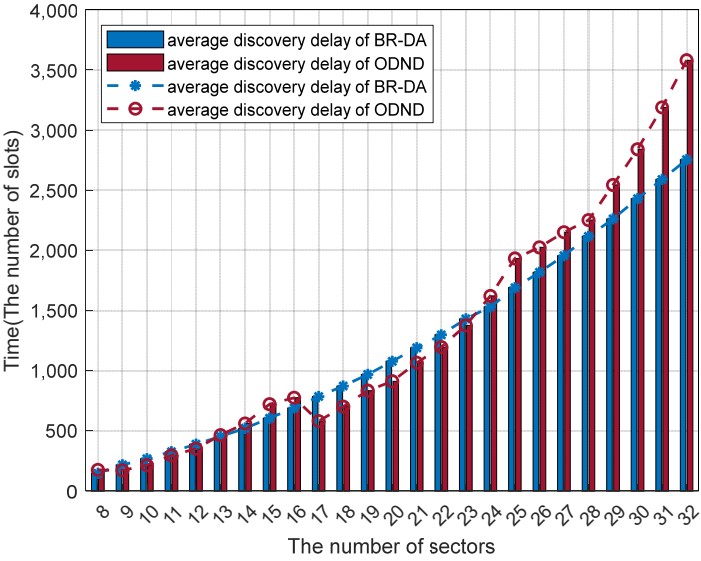

**Figure 26.** The average delay of the BR-DA and ODND protocols when ranging the number of sectors from 8 to 32.

### 4.2. Simulation on Network-Wide Neighbor Discovery

In order to further validate the neighbor discovery in the FANET, we conducted simulations in a network consisting of 100 randomly deployed UAVs within a 200 m × 200 m square area. The transmission range of UAVs varied from 25 m to 125 m, covering a wide range of practical scenarios. The transmission range and the number of sectors for each UAV in the FANET are the same.

We traced the worst-case and the average discovery delay of the BR-DA-FANET protocol and the ODND protocol. The worst-case discovery delay represents the maximum delay observed among 10,000 runs, while the average discovery delay is the average value over the same number of runs.

Figure 27 presents the worst-case discovery delay for different transmission ranges and different numbers of sectors. It clearly demonstrates that the worst-case discovery delay of the BR-DA-FANET protocol outperforms that of the ODND protocol. As the transmission range increases, the number of neighbors for each UAV also increases, resulting in an increase in the probability of worst-case delay.

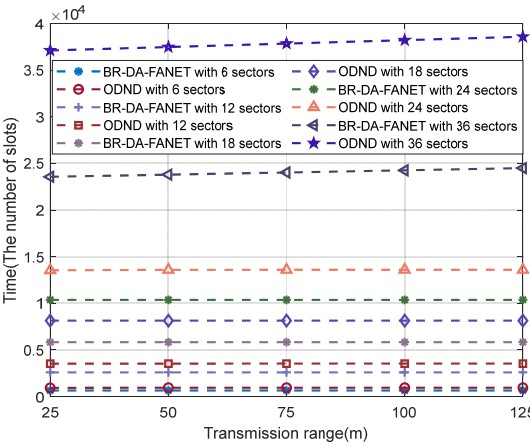

**Figure 27.** The worst-case discovery delay under different transmission ranges and different numbers of sectors.

The average discovery delay under different transmission ranges as well as different numbers of sectors is shown in Figure 28. Figure 28 shows that the average discovery delay of the BR-DA-FANET protocol is less than that of the ODND protocol. With increasing transmission range, the average delay is increasing, and the explanation is the same as the worst-case delay.

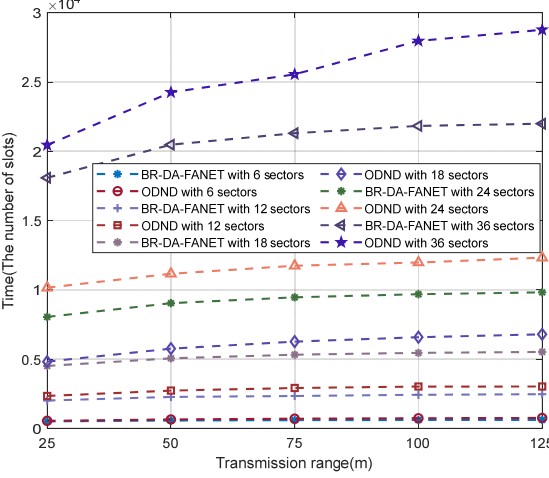

**Figure 28.** The average discovery delay under different transmission ranges and different numbers of sectors.

### 4.3. Simulation on Main Lobe Rendezvous Scheme Based on Location Prediction

Each UAV in a FANET can discover all neighbors according to the BR-DA-FANET protocol. As a UAV prepares to communicate with any neighbor, it needs to discover the neighbor quickly according to the ND-LP protocol. The random waypoint mobility model is adopted as the mobility model to mimic the movement of a flying UAV [36,37].

Once the initial access for the FANET is completed, each UAV discovers one of its neighbors 99 times. After neighbor discovery, each UAV moves in a random direction at a certain random speed for a period of time. The neighbor discovery delay is calculated as the average of 10,000 runs in our simulation.

The simulation result is shown in Figure 29. As shown, the delay for the first neighbor discovery of the ND-LP protocol is as large as that of the BR-DA protocol. However, from the second neighbor discovery onwards, the delay decreases quickly due to the location prediction based on the information obtained from the previous neighbor discovery.

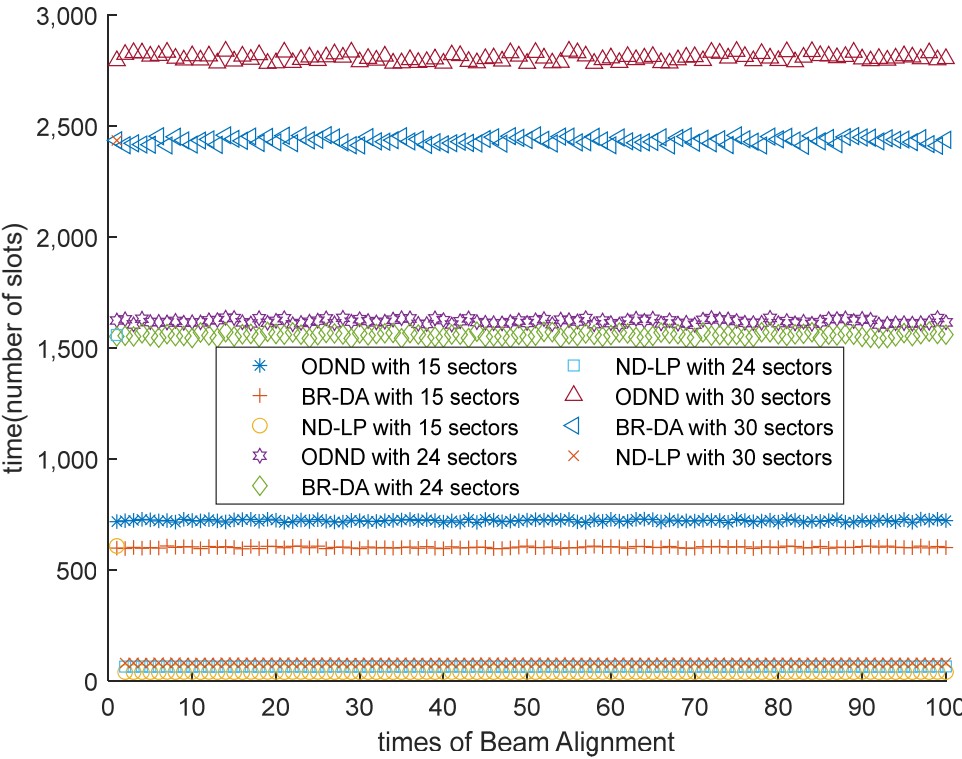

**Figure 29.** Discovery delay of continuous 100 times neighbor discovery under different numbers of sectors.

The communication range of the switched beam antenna model overtakes the omnidirectional antenna because of high gain. Therefore, the design of double-directional antennas can reduce the number of hops required in comparison to using the omnidirectional antenna. To validate this performance improvement, we conducted simulations. In our simulations, the gains of the switched beam antenna and phased array antenna are 10 dB and 20 dB, respectively. The gain of the omnidirectional antenna is 0 dB. According to the Friis transmission equation [31], the minimum number of hops required for the omnidirectional antenna to reach the maximum communication range of the phased array antenna is 10, while for the switched beam antenna, it is only 4 hops.

The communication range of the omnidirectional antenna is 10 m. Hence, the communication ranges of the switched beam antenna and phased array antenna are $10 * 10^{0.5}$ m and $10^2$ m, respectively. In our simulation scenario, there is a pair of randomly positioned UAVs with a distance ranging from 0 to 100 m between them. Several relay UAVs are placed between this pair to facilitate multihop communication. Both the omnidirectional antenna and switched beam antenna extend the communication range to outnumber the

distance between this pair of UAVs by multihops. To compare the performance of the omnidirectional antenna and the switched beam antenna, we measure the number of hops required to reach the distance between this pair of UAVs, as presented in Figure 30. Our simulation demonstrates that the switched beam antenna significantly reduces the number of hops compared to the omnidirectional antenna.

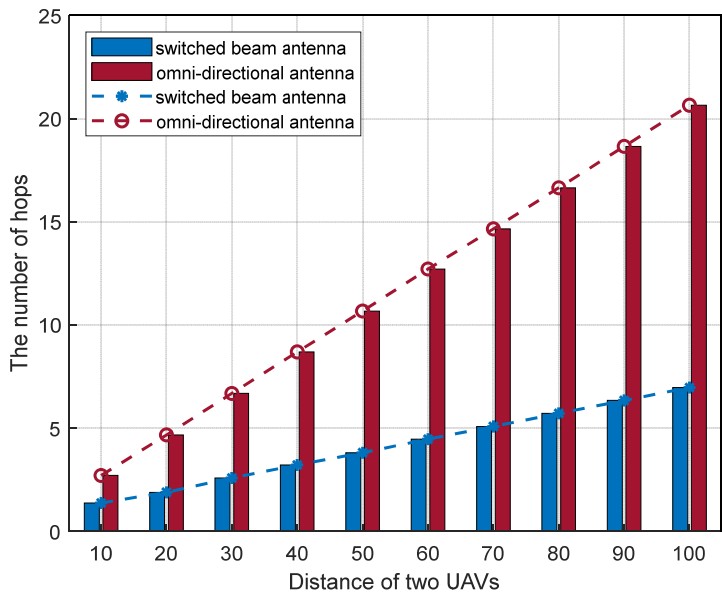

**Figure 30.** The number of hops for the omnidirectional antenna and switched beam antenna.

*4.4. Simulation for Method for Avoiding Communication Interruption Based on Location Prediction*

Communication interruptions can significantly increase the transmission delay. In this section, we present simulation results to demonstrate the effectiveness of the ACI-LP protocol in reducing transmission delay.

Our simulation scenario involves a pair of continuously moving UAVs that communicate with each other. The relevant parameters are shown in Table 3. Before transmitting each packet, a carrier sense mechanism is employed to assess the availability of channels, taking into account the presence of multiple channels. Without the ACI-LP protocol, the pair of UAVs would need to rediscover each other due to communication interruptions.

**Table 3.** The parameters for simulation of the ACI-LP protocol.

| Parameter | Value |
| :---: | :---: |
| $T_{CS}$ | 1 ms |
| $T_{TS2-RTS}$ | 1 ms |
| $T_{TS2-CTS}$ | 1 ms |
| $T_{Inf}$ | 5 ms |
| $T_{ACK}$ | 1 ms |
| $T_{TS1-RTS}$ | 1 ms |
| $T_{TS1-CTS}$ | 1 ms |
| The number of sectors | 30 |

The angle of the main lobe, speed of UAVs, and distance between the pair of UAVs influence the time of the main lobe rendezvous. Therefore, we investigate the influence of the time of main lobe rendezvous on transmission time.

The simulation results are shown in Figure 31. In the figure, T-mlr represents the time of main lobe rendezvous, while the time of all information indicates the ratio of transmitted information to the communication rate. The actual time of all information is the communication time considering communication interruptions.

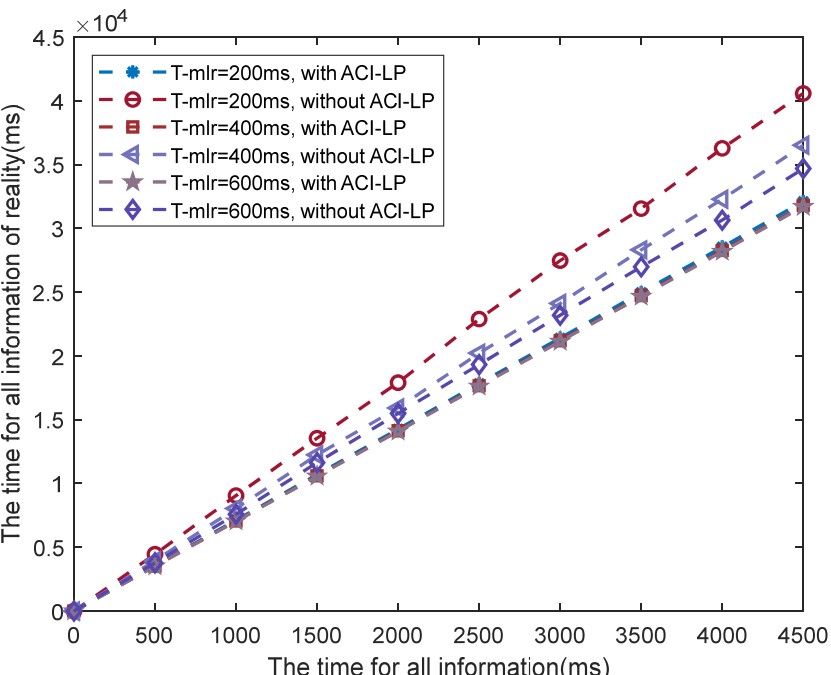

**Figure 31.** The simulation for an ACI-LP protocol.

Figure 31 clearly demonstrates that the ACI-LP protocol decreases the actual communication delay caused by interruptions.

## 5. Conclusions

In this paper, we introduce the FA-MMAC-DA protocol for a directional FANET with multiple channels to achieve fast neighbor discovery and avoid communication interruption. To address the challenge of neighbor discovery for a FANET under the circumstances of being without coordination or prior information, the BR-DA and BR-DA-FANET protocols are proposed to reach the theoretical supremum and achieve quick neighbor discovery during the initial access phase. Subsequently, we present the ND-LP and ACI-LP protocols to enable efficient neighbor discovery and mitigate communication interruptions. Through simulations, we demonstrate that the FA-MMAC-DA protocol outperforms existing approaches by reducing both neighbor discovery delay and communication delay. The protocol proposed in this paper is applicable to a dedicated communication directional ad hoc network with multiple nodes and multiple channels, such as collaborative reconnaissance with multiple UAVs, coordinated disaster relief efforts, and more. For instance, in scenarios where multiple UAVs engage in collaborative communication to facilitate coordinated reconnaissance in disaster-stricken areas, communication among UAVs is established using the FANET. This protocol can be employed to achieve rapid neighbor discovery within the FANET network, thereby reducing neighbor discovery delay and transmission delay.

**Author Contributions:** This research was accomplished by all the authors: S.L., H.Z., J.Z., H.W., J.W. (Jibo Wei) and J.W. (Junfang Wang) conceived the idea, performed the analysis, and designed the scheme; S.L., J.Z. and H.W. conducted the numerical simulations; S.L., H.Z., J.W. (Jibo Wei) and J.W. (Junfang Wang) cowrote the manuscript. All authors have read and agreed to the published version of the manuscript.

**Funding:** This research was funded by the National Natural Science Foundation of China (grants: 62001483 and 61931020) and the project was supported by the Provincial Natural Science Foundation of Hunan (grant: 2022JJ10068).

**Data Availability Statement:** Data is contained within the article.

**Conflicts of Interest:** Shijie Liang and Junfang Wang are employed by the 54th Research Institute of China Electronics Technology Group Corporation. The remaining authors declare that the research was conducted in the absence of any commercial or financial relationships that could be construed as a potential conflict of interest.

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
