# Peer review of "A Multichannel MAC Protocol without Coordination or Prior Information for Directional Flying Ad hoc Networks"

_drones, doi:10.3390/drones7120691_

Round 1
Reviewer 1 Report
Comments and Suggestions for Authors
Please see the attachment.

I consider the quality of english language is adequate for a research article. However, there are some minor errors throughout the paper.
Reviewer 2 Report
Comments and Suggestions for Authors
Not being an expert on this particular topic, I think that the manuscript is well written and the results of interest. I would recommend its publication after having answered the following questions:
Between Fig.3 and 4: Isn't there 6 and not 5 pairs of UAV communicating?
Above Fig.7: I cannot find Sec.III C. referred there. It also appears at other locations in the manuscript.
In Eqs.(5), (6): why do you need the integer Z in the indices of the sums?
Could the authors comment on the applicability of their methods with UAVs that have different dynamics, e.g. with motion that depends on the neighbors?
Reviewer 3 Report
Comments and Suggestions for Authors
This paper addresses an important topic: achieving neighbor discovery for directional FANETs with multiple channels, without coordination or prior information. This is a challenging task due to communication interruptions caused by the high mobility of UAVs. The goal is to minimize delay in achieving neighbor discovery and avoid communication interruptions in scenarios with directional antennae and multiple channels. The authors introduce the FA-MMAC-DA, which consists of several protocols: BR-DA and BR-DA-FANET, ND-LP, and ACI-LP.
The paper is well-written. The authors first provide a network model, an antenna model, and a multichannel directional FANET model before defining the problem. The theoretical background behind the proposed blind rendezvous algorithms (BR-DA and BR-DA-FANET, ND-LP, and ACI-LP) is solid. The results are well-presented and discussed. A comparison with recent works is provided, and the proposed approach shows significant improvement.
However, there are some areas where the paper could be improved:
-
Line 195: The authors should double-check the statement, "In slot 1, there are five pairs of UAVs engaging in communication." It appears that it should be six pairs, not five.
-
For better readability, it is recommended to create spaces between figures and between figures and text. Figures 18 and 19 are too close together. The same issue exists with Figure 21 and Equation 22.
-
Figures 2 and 4: The visibility of labels A and B in Figure 2 should be improved. Additionally, the labels in Figure 4 should be updated for clarity.
-
Abbreviations and Variable Descriptions: Authors should consider adding a table for abbreviations and another one for variable and parameter descriptions. This will enhance the readability of the document and help readers easily understand the terms used.
-
Assumptions Section: It would be beneficial to add a section related to assumptions to provide a clearer understanding of the network, antenna, and multichannel directional FANET models. This can help readers better contextualize the research.
-
Comparison with ODND: The authors should provide a rationale for comparing their results with only ODND from 2017. Explaining the reason for this choice will add depth to the discussion.
-
Figure 25 Discussion: The discussion on Figure 25 could be extended, especially for cases where the average discovery delay of BR-DA is greater than that of ODND. Providing insights into these cases can enhance the paper's completeness.
-
Future directions: The authors might consider adding a section discussing the potential future directions or applications of their work. This can provide readers with insights into the broader implications of the research.
